# The Value of Magnetic Resonance Diffusion-Weighted Imaging and Dynamic Contrast Enhancement in the Diagnosis and Prognosis of Treatment Response in Patients with Epithelial Serous Ovarian Cancer

**DOI:** 10.3390/cancers14102464

**Published:** 2022-05-17

**Authors:** Pawel Derlatka, Laretta Grabowska-Derlatka, Marta Halaburda-Rola, Wojciech Szeszkowski, Krzysztof Czajkowski

**Affiliations:** 1Second Department Obstetrics and Gynecology, Medical University of Warsaw, Karowa 2 St., 00-315 Warsaw, Poland; krzysztof.czajkowski@wum.edu.pl; 2Second Department of Clinical Radiology, Medical University of Warsaw, Banacha 1a St., 02-097 Warsaw, Poland; mhalaburda1989@gmail.com (M.H.-R.); wszeszkowski@wum.edu.pl (W.S.)

**Keywords:** serous epithelial ovarian cancer, magnetic resonance diffusion-weighted imaging (DWI), magnetic resonance dynamic contrast enhancement (DCE), progression free survival (PFS), prognostic factors

## Abstract

**Simple Summary:**

Epithelial ovarian cancer is one of the greatest challenges for a gynecologist and oncologist both in terms of diagnosis and treatment. Modern imaging techniques such as DWI or DCE MRI allow for better planning of the treatment strategy. This is related not only to a more precise localization of lesions, but also to the relationship between the values of DWI and DCE parameters and specific histological types of ovarian cancer. In our study, we demonstrated the previously suggested relationships between the values of DWI parameters and the types of ovarian cancer. We described the relationship with the results of immunohistochemical tests. We also showed a correlation between DWI and DCE values with time to relapse. We have made an attempt to describe such correlations in the group of patients treated with bevacizumab.

**Abstract:**

Background. The aim of our study was to describe the selected parameters of diffusion-weighted imaging (DWI) and perfusion dynamic contrast enhancement (DCE) MRI in primary tumors in patients with serous epithelial ovarian cancer (EOC), as well as in disease course prognosis and treatment response, including bevacizumab maintenance therapy. Materials and Methods. In total, 55 patients with primary serous EOC were enrolled in the study. All patients underwent MR imaging using a 1.5 T clinical whole-body MR system in preoperative DWI and DCE MRI selected parameters: apparent diffusion coefficients (ADC), time to peek (TTP) and perfusion maximum enhancement (Perf. Max. En.) were measured. The data were compared with histopathological and immunochemistry results (with Ki67 and VEGF expression) and clinical outcomes. Results. Higher mean ADC values were found in low-grade EOC compared to high-grade EOC: 1151.27 vs. 894,918 (*p* < 0.0001). A negative correlation was found between ADC and Ki67 expression (*p* = 0.027), and between ADC and VEGF expression (*p* = 0.042). There was a negative correlation between TTP and PFS (*p* = 0.0019) and Perf. Max. En. and PSF (*p* = 0.003). In the Kaplan–Meier analysis (log rank), a longer PFS was found in patients with ADC values greater than the median; *p* = 0.046. The Kaplan–Meier analysis showed a longer PFS (*p* = 0.0126) in a group with TTP below the mean value for this parameter in patients who received maintenance treatment with bevacizumab. Conclusions. The described relationships between PFS and DCE and DWI allow us to hope to include these parameters in the group of EOC prognostic factors. This aspect seems to be of particular interest in the case of the association of PFS with DCE values in the group of patients treated with bevacizumab.

## 1. Introduction

Epithelial ovarian cancer (EOC) is the fifth most common cancer in women. It is also the fourth leading cause of death from cancer because of the lack of discernible symptoms and effective screening tools [1,2]. Tumor prognosis depends on the use of optimal cytoreductive surgery and adjuvant platinum-based chemotherapy [3,4].

Surgical outcome in EOC is usually classified according to the amount of postoperative residual tumor. A complete resection is regarded if no macroscopically visible tumor is left. If any visible tumor remains after surgery, it is classified according to its largest diameter. Operations that ended up with residuals up to the 10 mm largest diameter had been formerly classified as optimal debulking, whereas those resulting in any larger residual tumor had been defined as suboptimal debulking [5,6].

Following the publication of the results of two III-phase clinical trials (GOG 218 and ICON 7), the adjuvant treatment of advanced or high-risk early stage EOC is as follows: six 3-weekly cycles of intravenous carboplatin (AUC 5 or 6) and paclitaxel (175 mg/m^2^ of body surface area), with maintenance intravenous bevacizumab in a dose of 7.5 mg/kg of bodyweight continued for twelve further 3-weekly cycles (ICON 7) or 15 mg/kg of bodyweight during sixteen 3-weekly cycles (GOG 218) [7,8,9].

Transvaginal ultrasonography (TVUS) is the initial modality for investigating ovarian tumors, and the International Ovarian Tumor Analysis Group guideline can be used to estimate the malignancy risks of ovarian tumors [10]. According to the guidelines of the European Society of Uro-Genital Radiology (ESUR), the imaging modality of choice for the preoperative evaluation of such patients is abdomino-pelvic and chest computed tomography (CT) [11]. However, the CT in some cases is not able to indicate very small and diffuse peritoneal implants or bowel wall or mesentery infiltrations. In the pretreatment diagnosis of EOC, magnetic resonance imaging (MRI) can yield more information than CT. Compared to CT, MRI diffusion weighted imaging (DWI) has shown promise in tumor staging, predicting the aggressiveness of the tumor and clinical outcome [12,13].

DWI, in combination with apparent diffusion coefficients (ADC), bring new possibilities in the imaging of EOC, particularly in diagnosing intraperitoneal implants. According to recent studies, diffusion restriction is higher in intraperitoneal implants than in primary tumors [14]. Other studies confirm that ADC values correlate with vascular endothelial growth factor (VEGF), whose expression is higher in intraperitoneal implants than in primary tumors. There is also a reported inverse correlation between ADC values and Ki67 protein, the proliferation marker [15].

Dynamic contrast-enhanced (DCE) MRI is used to improve the diagnostic accuracy of conventional MRI. Most DCE-MRI studies of ovarian tumors have targeted differentiating among benign, borderline and malignant lesions [16]. There is some information that DCE parameters may be useful in the differentiation between highly and low aggressive EOC [17]. Moreover, some studies proposed the application of perfusion-MRI as a prognostic study in EOC [17]. There are no unequivocal data on the relationship between the results of DWI and perfusion MRI in the primary tumor and disease progression. 

The aim of our study is an attempt to describe the parameters of DWI and DCE MRI in primary tumors in patients with serous EOC. Additionally, we analyzed selected parameters of DWI and DCE of the primary tumor in early and advanced disease as well as in disease prognosis.

## 2. Material and Methods

### 2.1. Study Protocol and Patients Population

A single-center prospective study was conducted at the Medical University of Warsaw in the 2nd Department of Obstetrics and Gynecology and the 2nd Department of Clinical Radiology. The inclusion criteria for the study were clinical suspicion of ovarian cancer in CT or TVUS. The exclusion criteria were contraindications to MRI with gadolinium contrast, the current therapy of coexisting neoplasms, starting EOC chemotherapy before performing MRI and surgery outside our center.

The study included 55 women aged 30–78 years with primary serous EOC diagnosed in the final histopathological examination. The type and histological differentiation were assessed according to the WHO criteria of 2014, and the serous EOC was classified into low-grade (LG) and high-grade (HG) EOC. The advancement of the disease was assessed according to the FIGO criteria (International Federation of Obstetricians and Gynecologists).

### 2.2. Treatment Protocol

First-line treatment consisted of primary cytoreduction followed by chemotherapy. In patients disqualified from primary cytoreduction, an exploratory laparoscopy was performed to establish the histopathological diagnosis, followed by the neoadjuvant chemotherapy. Systemic treatment was continued after the postponed cytoreduction.

The adjuvant treatment consisted of six courses of intravenous carboplatin (AUC 5 or 6) and 175 mg/m^2^ body surface area paclitaxel administered every three weeks. According to the results of the ICON study, 7 patients at high risk of relapse received maintenance treatment with bevacizumab at a dose of 7.5 mg/kg every three weeks for a total of 18 courses or until progression. Neoadjuvant chemotherapy consisted of the administration of 3 courses according to the above-mentioned scheme. After delayed cytoreduction, chemotherapy was continued for up to 6 or 8 courses, and in high-risk patients, bevacizumab was administered for a total of 18 courses or until progression. Patient characteristics and clinical-histopathological data are presented in Table 1.

### 2.3. MRI Protocol

All patients underwent MR imaging using a 1.5 T clinical whole-body MR system (MAGNETOM Avanto; Siemens AG, Erlangen, Germany).

The MRI protocol for the detection of the pelvic and abdominal lesions contained turbo spin-echo (TSE), T2-weighted images (T2 w), fat-suppressed T2-weighted (fsT2 w), turbo inversion recovery magnitude (TIRM), diffusion-weighted echo planar imaging (DW-EPI) and pre- and postcontrast dynamic T1-weighted gradient echo (3D T1 GRE) sequences. The details of the applied parameters of MR imaging are presented in Table 2. Axial DW images were acquired using the same multi-slice EPI sequence for all patients 30 × 6 mm slices (pelvic part); 360 × 360 mm FoV; TR = 4250 ms; TE = 73 ms; with diffusion weightings of 0, 50, 500, 1000 and 1500 mm^2^/s. These parameters are shown in Table 2. Motion correction was completed automatically. 

Two radiologists experienced in pelvic MRI, and blinded to the histological information, documented the character of the adnexal masses (one board specialist with more than 15 years of experience and a specialist with a European Diploma in Radiology certificate). Regions of interest (ROI) were drawn on the ADC maps and all b values DWI outlined in Multimodality Workplace Station (GE AW Serwer 3.2 ext 4.0, Volume Viewer 16.0 Ext.2 Ready View, 42655 Sollingen, Germany).

On all DWI (with b values of 0, 50, 500, 1000, 1500 mm^2^/s), ROI contained the small circle with a diameter 5–6 mm which was placed on the solid part of the primary tumor, avoiding the partial volume effect, areas of necrosis and artifacts. ROI were replicated from the DW image to the corresponding ADC map and the measurement on the ADC map was recorded. T1WI (non-contrast and contrast enhanced), and DCE sequence parameters for the dynamic analysis are presented in Table 2. ROI were drawn on enhancement DCE images and replicated to DCE parameter maps. During DCE image acquisition, non-contrast images were acquired first, followed by contrast agent administration and continued image acquisition. Time to peek (TTP) and perfusion maximum enhancement (Perf. Max. En.) were measured. In all patients, Gadobutrol (Gadovist, Bayer Schering, Berlin, Germany) was administered, as a bolus dose of 0.1 mmol/kg, immediately followed by a bolus dose of 20 mL of physiological saline (NaCl 0.9%).

DCE parameter maps were generated automatically using Workplace Station.

### 2.4. Immunohistochemistry

The study included samples obtained from the primary tumor prior to the initiation of chemotherapy. In patients undergoing neoadjuvant chemotherapy, the material was obtained from tissue collected during laparoscopy. The tissue was embedded in paraffin and then cut into 5 µm (micrometer) thick sections. A histopathological examination was performed after staining with hematoxylin and eosin. In the Ki67 immunohistological study, the En Vission FLEX Mini Kit High pH was used, while in the VEGF study, the DAKO Monoclonal Mouse Anti-Human VEGF Clone VG1, 1:50 was used. The expression of Ki67 was assessed in the cell nucleus and VEGF in the epithelium and stroma. Ki67 was determined in all 55 patients, and VEGF in 51 patients. The result was reported as the percentage of cells showing staining.

### 2.5. Statistical Analysis

Dell Statistica (data analysis software system, version 13.1) and MedCalc (ver. 20.014, MedCalc Software Ltd., Acacialaan, 8400 Ostend, Belgium) were used for the statistical analysis. All continuous variables were assessed for normality using a one-sample Kolmogorov–Smirnov test, and the data were expressed as the mean standard deviation or median. The parametric *T*-tests for independent groups were used for testing the significance of differences between mean values because of normal distribution. All correlations were analyzed using a linear model with the Pearson correlation coefficient. For the survival analyses, imaging parameters (ADC, TTP, Perf. Max. Enh.) were dichotomized using the mean values as a cut-off. Recurrence-free survival (RFS) was defined as the interval between the date of surgery and the date of identified recurrence, and overall survival (OS) as the interval between the date of surgery and the date of death or the end of follow-up. The Kaplan–Meier method (log rank) was used for the univariate survival analysis. A *p*-value of <0.05 was considered statistically significant.

## 3. Results

The inclusion criteria for the study were met by 55 patients, whose median age at diagnosis was 57 years (30–78). There were 74.5% of patients in FIGO Grades III and IV (*n* = 41).

### 3.1. Primary Tumor

All studies managed to visualize the primary tumor in which the ROI was located. The median of the greatest size of the primary tumor was 78 mm (range 60–139). LG EOC was diagnosed in 16 patients, and HG EOC in 39 patients (Figure 1, Figure 2 and Figure 3). 

A very high agreement was obtained both in the results of the two ADC measurements obtained by each radiologist, and in the comparison of the mean measurements between the radiologists. The intraclass correlation coefficient (ICC) for radiologist A’s mean measurement was 0.966. The ICC for radiologist B’s mean measurement was 0.955. Concordance between radiologists A and B for the mean of the first measurement was ICC = 0.932, and for the mean of the second measurement, ICC = 0.916 (Figure 4).

A significantly higher mean of ADC values was found in low-grade EOC tumors compared to high-grade EOC tumors: 1151.27 vs. 894,918 (*p* < 0.0001). No differences were found in TTP (*p* = 0.87) and Perf. Max. En. (*p* = 0.43) for these histopathological diagnoses (Table 3).

### 3.2. MRI DWI and DCE Parameters and Immunohistochemistry

Examples of Ki67 and VEGF immunohistochemical staining are provided in Figure 5 and Figure 6.

A significant negative correlation was found between ADC values and Ki67 expression; *p* = 0.027, r = −0.298 (Figure 7), and a negative correlation (*p* = 0.042, r = −0.285) between ADC and VEGF expression in the primary tumor (Figure 8). No correlation was found between TTP, Perf. Max. En. value with Ki67 and VEGF (Table 4).

There were no significant differences between ADC values (*p* = 0.289), TTP (*p* = 0.230) and Perf. Max. En. (*p* = 0.107) in the primary tumor in the extragonadal disease progression (grade I vs. stages II–IV).

### 3.3. Relapse of the Disease

In total, 34 patients (all FIGO grades) had a recurrence of the disease. The mean PFS was 17.6 months (range 0–40). There was a significant negative correlation between TTP and PFS values; *p* = 0.0019, r = −0.51 and between Perf. Max. En. and PSF; *p* = 0.003 and r = −0.49. No significant relationship was found between ADC and PFS (*p* = 0.836) (Table 5, Figure 9 and Figure 10).

No correlation was found between ADC (*p* = 0.12), TTP (*p* = 0.55) and Perf. Max. En. (*p* = 0.26) and OS.

In the Kaplan–Meier analysis (log rank), a significantly longer PFS was found in the group of patients with ADC values greater than the median; *p* = 0.046.

No such correlation was found in the Kaplan–Meier analysis between TTP; *p* < 0.19 and Perf. Max. En.; *p* < 0.39 (Figure 11).

Survival curves were also analyzed in the group of patients who received maintenance treatment with bevacizumab. A Kaplan–Meier analysis showed a longer PFS in patients with TTP values below the mean value for this parameter; *p* = 0.0126 (Figure 12). In the case of Perf. Max. En. and ADC, no such correlation was found.

## 4. Discussion

Our study involving 55 patients showed significant correlations between ADC and the results of histopathological and immunohistochemical tests (Ki67, VEGF) of serous EOC, confirming its significance in predicting the course of the disease. We also showed that DCE parameters such as TTP and Max. Perf. En. also correlate with PFS. We were probably the first to analyze and describe the correlation between DCE parameters and PFS in patients receiving maintenance treatment with bevacizumab.

EOC is the biggest problem faced by people who treat cancer of the female genital organs. It is usually diagnosed in advanced stages III and IV. It especially concerns the serous type studied by us. According to the National Cancer Institute data, high-grade serous ovarian cancer is diagnosed in 51% in stage III and in 29% in stage IV according to FIGO [18].

The standard of treatment for EOC is primary optimal or complete cytoreduction followed by platinum-based chemotherapy. The 2019 ESMO ESGO consensus confirmed the role of cytoreductive surgery as a prognostic factor in EOC [19]. Patients operated on without leaving macroscopic disease or with macroscopic disease up to 10 mm have a better prognosis than patients with a left tumor larger than 10 mm [20]. In patients who cannot perform primary optimal cytoreduction, treatment is started with exploratory laparoscopy and neoadjuvant chemotherapy [21]. Hence, the huge role of preoperative imaging tests not only in determining the advancement of the disease, but also in qualifying for an appropriate treatment method [22]. According to the recommendations of the European Society of Urogenital Radiology (ESUR) from 2010, the recommended imaging modality for the management and initial preoperative staging is CT of the chest, abdominal cavity and pelvis [11]. However, we know that a CT examination has a number of limitations, especially in the diagnosis of intraperitoneal dissemination, especially in the case of small lesions, without the presence of ascites [23,24]. It seems that MRI can bring new possibilities both in the differentiation of ovarian neoplasms as well as in the prognosis of the course of the disease. Previous reports have found that ADC values correlate with established immunohistochemical prognostic factors for ovarian cancer such as the proliferation marker Ki67. In our research, we confirmed the negative correlations between Ki67 and ADC (r= −0.2981, *p* = 0.027). Lower ADC values and higher Ki67 values correspond to poorly differentiated EOC, which is associated with a worse prognosis. Thus, we confirmed the results obtained in the research by Lindgren et al. on EOC [15]. Similar negative correlations of ADC and Ki67 are also found in ductal breast cancer research (r = −0.717 do r = −410, *p* < 0.001) [25], prostate cancer (r = −0.332, *p* < 0.05) [26] and rectal cancer (r = −0.555, *p* < 0.001) [27]. This dependence is indirectly confirmed by the correlation of ADC with two types of EOC: low-grade and high-grade (type I and II). Greater diffusion restriction and thus lower mean ADC values were recorded in high-grade EOC tumors, i.e., tumors with lower differentiation and with higher aggressiveness [28]. Our studies confirmed this observation, and the negative correlation of ADC with low-grade and high-grade EOC was significant (*p* < 0.0001). Currently, the MRI gives more and more possibilities to distinguish features such as FS-T2WI, DWI, CE-T1WI and DCE, which allow for the differentiation of two types of EOC [29,30].

VEGF is one of the most important cytokines responsible for angiogenesis in EOC. By binding to a cellular receptor, it is involved in the formation of new tumor vessels [31]. However, one of the first studies on VEGF in ovarian cancer found that in patients with advanced EOC, intense VEGF immunostaining was more often detected in peritoneal metastases than in primary tumors. VEGF immunostaining in primary as well as in metastatic lesions correlated neither with the response to chemotherapy nor with the clinical outcome. Therefore, the detection of VEGF in tissue samples failed to have a predictive or prognostic relevance for patients with advanced OC [32]. In the other study from that time, the authors concluded that VEGF-C, VEGF-D and VEGFR-3 play an important role in lymphatic spread and intraperitoneal tumor development in OC [33].

After about ten years, it was proven that VEGF inhibitory factor and its receptor is the anti-VEGF antibody-bevacizumab, used in the maintenance therapy of advanced EOC [34,35]. Its effectiveness was confirmed by the ICON 7 and GOG 2018 studies mentioned in the introduction [7,8,9,36]. In our study, the correlation between ADC and VEGF protein expression in the primary tumor was negative (r = −0.2858, *p* = 0.04). The result differed from that obtained by Lindgren in one of the earlier studies, which found no correlation between ADC and VEGF in the primary tumor. On the other hand, the negative correlation of ADC with the three receptor types VEGFr-1 (r = 0.838, *p* = 0.001), VEGFr-2 (r = 0.764, *p* = 0.006), VEGFr-3 (r = 0.627, *p* = 0.039) and VEGFr-mRNA was confirmed (r = 0.855, *p* = 0.001) in intraperitoneal dissemination [15]. Similar to ours, negative correlations of ADC with VEGF protein in the primary tumor have been shown in studies on prostate cancer (r = −0.714, *p* = 0.005) [26] and in rectal cancer (r = −0.290, *p* = 0.005) [27].

When analyzing the recurrent disease, we showed an inverse correlation between PFS and the values of TTP (*p* = 0.0019) and Perf. Max. En. (*p* = 0.003) in the primary tumor. Higher DCE values were associated with a shorter time to relapse. In the Kaplan–Meier analysis for the entire study group, we found differences in probable time to relapse in groups with ADC values above and below the mean for this parameter. Higher ADC values were associated with longer survival (*p* = 0.04). Perhaps this correlation is explained by the relationship between higher ADC values and a better differentiated neoplasm and a lower percentage of Ki67. However, the differences between the probable PFS length and the TTP and Perf. Max. En. values above and below the average were not confirmed.

Lindgren, in a study from 2019, confirmed the difference in PFS curves for other DCE parameters, such as contrast agent distribution volume (Ve) and plasma volume (Vp). For TTP, it showed longer PFS in the group where the value of this parameter was greater than the median (the opposite was true for Ve and Vp) [17]. Our analysis of the survival curves in the group of patients who received bevacizumab maintenance treatment seems interesting. The Kaplan–Meier analysis showed a longer PFS for the group with TTP values lower than the mean for this parameter; *p* = 0.0126. We did not obtain such a correlation in the case of ADC, although a negative correlation of ADC with the VEGF protein was previously shown. Our study seems to open up the topic of the correlation of DWI and DCE parameters with survival curves in the group of patients receiving maintenance treatment with bevacizumab. This topic requires further analysis.

Our work has several limitations. The first is the single-center nature of the study. The second limitation is the analysis of patients with serous EOC. It is true that this type accounts for over 75% of cases of this cancer, but in clinical practice we will encounter other types of EOC. The obtained parameters may then differ from those described in the study. However, narrowing the group to the serous type made it possible to standardize the study group. The third limitation is the small number of patients treated with bevacizumab. However, this form of maintenance treatment is used only in selected patients.

## 5. Conclusions

The correlation of DWI parameters with markers of proliferation (Ki67) and factors influencing angiogenesis such as VEGF in the tumor, as well as the significant correlation of ADC values with the EOC type (low-grade vs. high-grade), make the MRI an excellent tool in the diagnosis of serous ovarian cancer. The described correlations between PFS and DCE and DWI allow us to hope to include parameters such as TTP, Perf. Max. En. or ADC in the group of prognostic factors of EOC. These parameters seem to be of particular interest in the association of PFS with DCE values in the group of patients treated with bevacizumab. However, it requires further research.

## Figures and Tables

**Figure 1 cancers-14-02464-f001:**
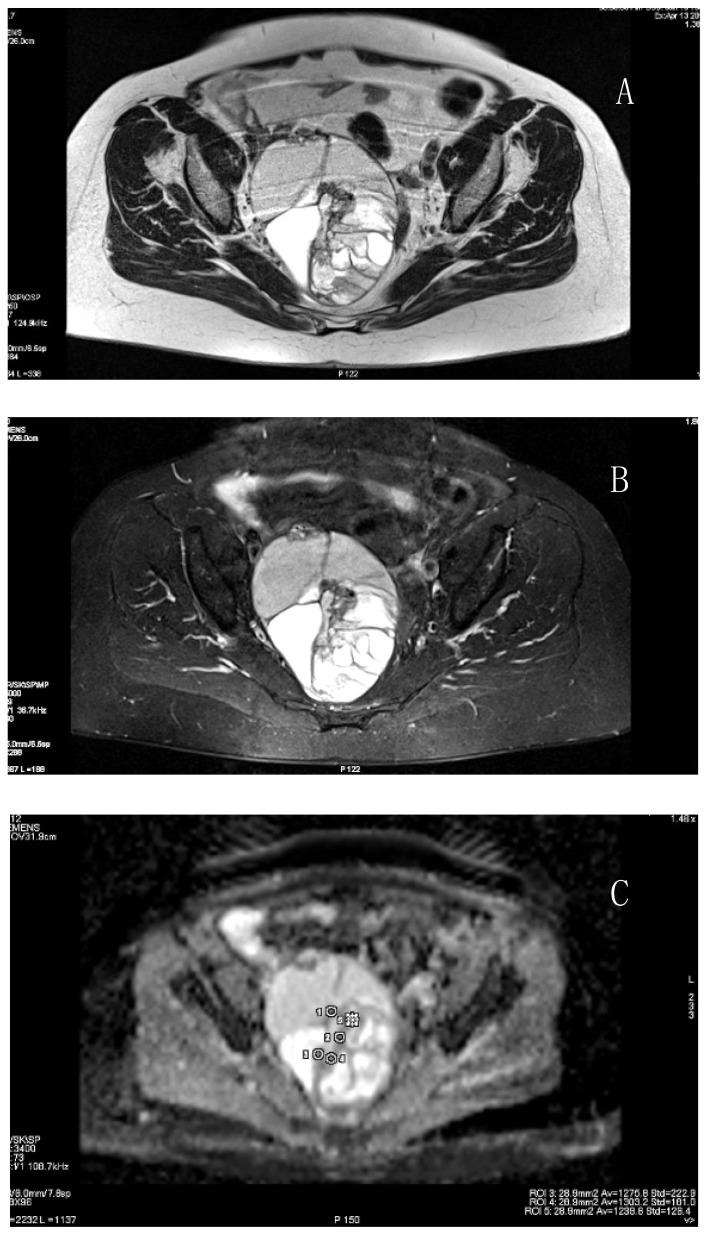
Images from 77 year-old women with low-grade serous ovarian cancer. (**A**): A large primary ovarian multicystic tumor on T2-weighted; (**B**): on T2 STIR; (**C**): Diffusion-ADC maps. Small ROI is placed on a region appearing to be the most enhancing solid part of the tumor.

**Figure 2 cancers-14-02464-f002:**
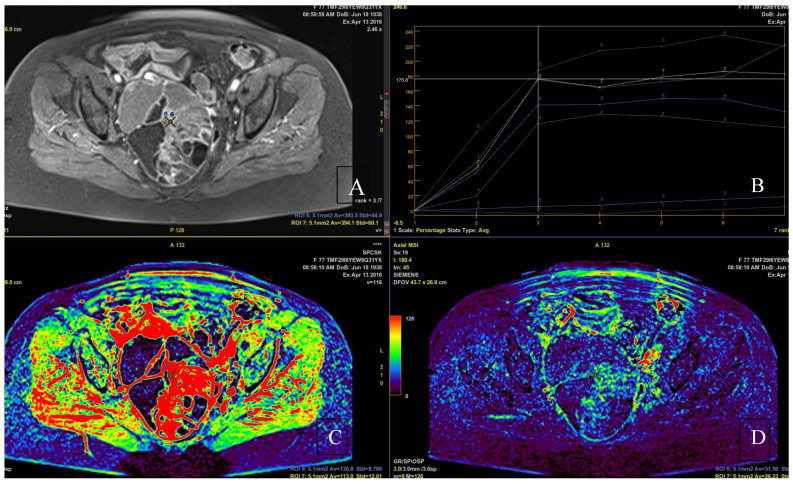
Images from the same 77 year-old women with low-grade serous ovarian cancer. (**A**,**C**,**D**) Dynamic contrast enhancement. (**B**) Contrast enhancement curves.

**Figure 3 cancers-14-02464-f003:**
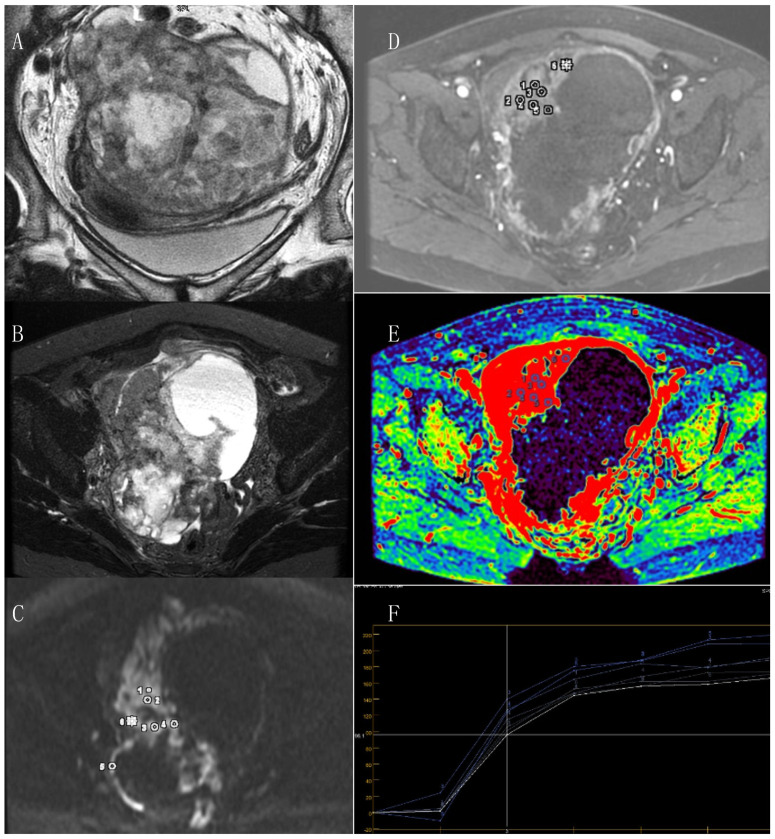
Images from 64-year-old women with high-grade serous ovarian cancer. (**A**): A large primary ovarian cancer cystic and solid part on T2-weighted; (**B**): T2 STIR; (**C**): DWI (b1500); (**D**–**F**) dynamic contrast enhancement small ROI is placed on a region appearing to be the most enhancing solid part of the tumor.

**Figure 4 cancers-14-02464-f004:**
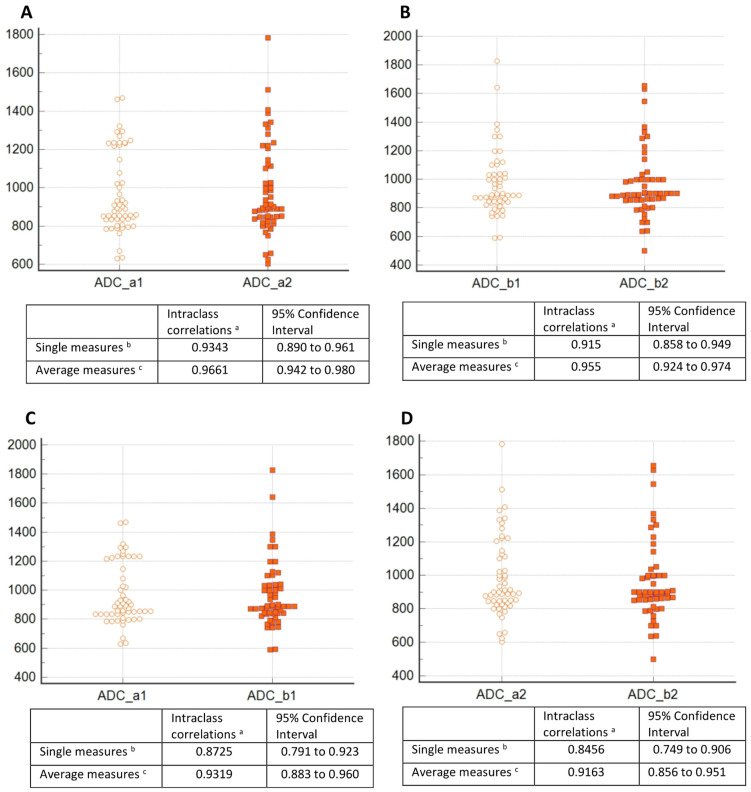
The interobserver agreement of the ADC measurements. A-concordance of the two measurements of the radiologist (**A**); B-agreement of the two measurements of the radiologist (**B**); (**C**) agreement of the first measurement between the radiologists (**A**) and (**B**); (**D**) concordance of the second measurement between the radiologists (**A**) and (**B**); ^a^ the degree of absolute agreement among measurements; ^b^ estimates the reliability of single ratings; ^c^ estimates the reliability of averages of k ratings.

**Figure 5 cancers-14-02464-f005:**
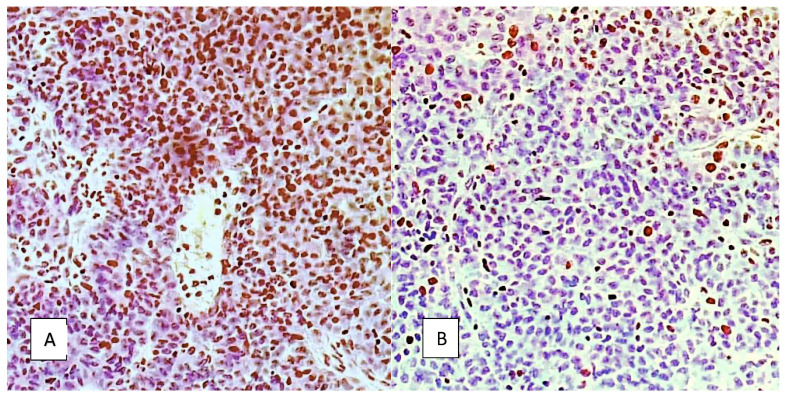
Ki67 staining of the nucleus in high-grade serous EOC, magnification 20×. (**A**) High expression, >90% positive stain cell nuclei; (**B**) low expression, 10% stain cell nuclei.

**Figure 6 cancers-14-02464-f006:**
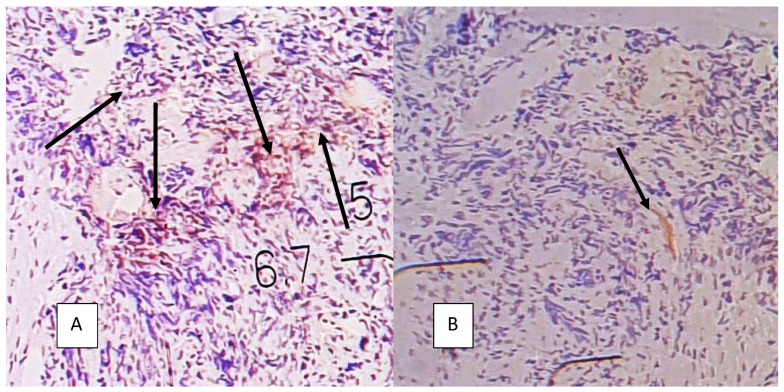
VEGF staining in high-grade serous EOC, magnification 20×. (**A**) High expression: many areas with stained cytoplasm are visible (arrows), (**B**) low expression: single area with stained cells (single arrow).

**Figure 7 cancers-14-02464-f007:**
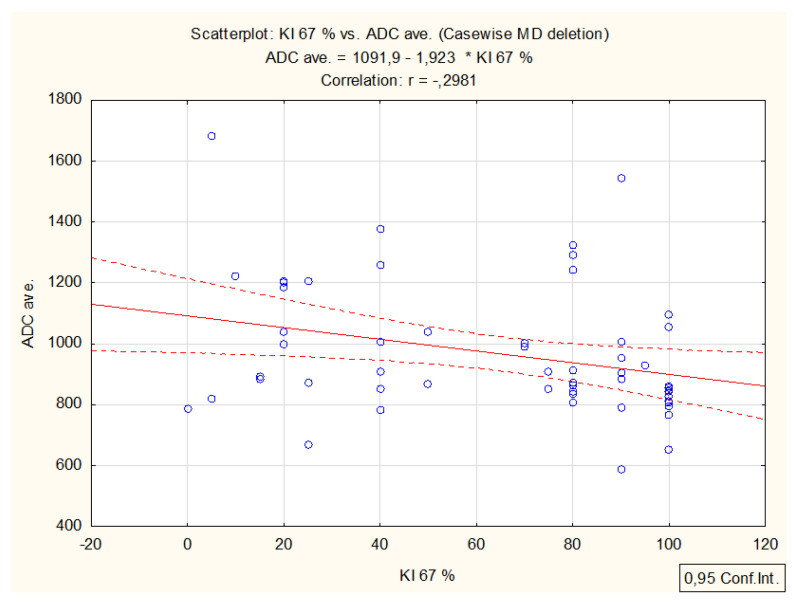
Correlation between Ki67 expression and mean ADC (*p* = 0.027).

**Figure 8 cancers-14-02464-f008:**
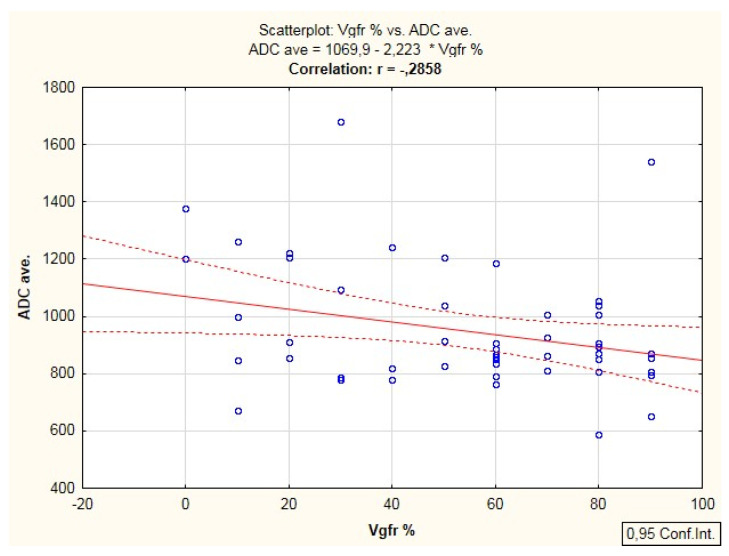
Correlation between VEGF expression and mean ADC (*p* = 0.042).

**Figure 9 cancers-14-02464-f009:**
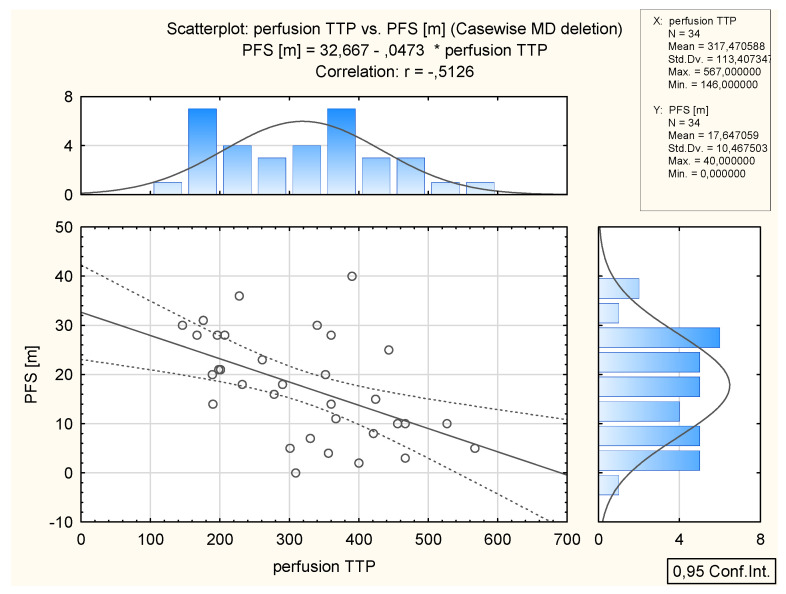
Inverse correlation between TTP and PFS (months) in 34 patients who had relapsed. Above: the distribution of TTP values; on the right: distribution of PFS (months).

**Figure 10 cancers-14-02464-f010:**
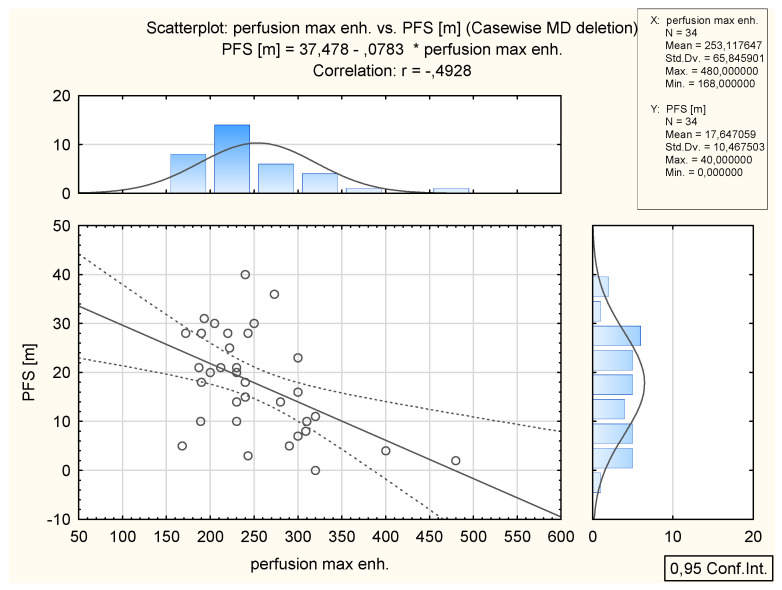
Inverse correlation between the values of Perf. Max. En. and PFS (months) in 34 patients who relapsed. Above: the distribution of Perf. Max. En. values; on the right: distribution of PFS (months).

**Figure 11 cancers-14-02464-f011:**
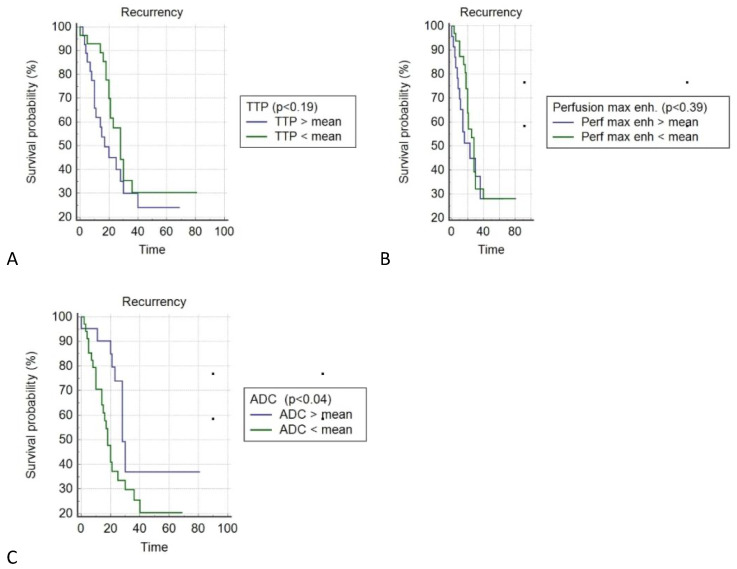
Kaplan–Meier (log rank) analysis of PFS for values below and above the median Perf. TTP; ns (**A**), Perf. Max. En.; ns (**B**) and ADC, *p* = 0.046 (**C**).

**Figure 12 cancers-14-02464-f012:**
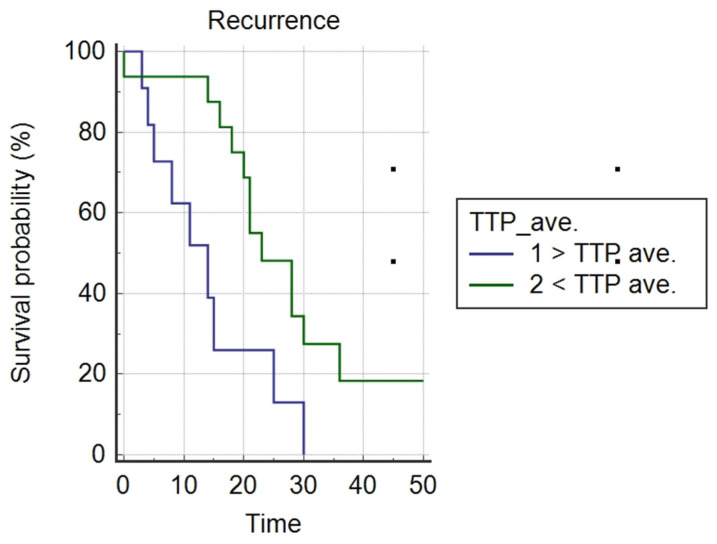
Kaplan–Meier (log rank) analysis of PFS for values below and above the mean TTP among patients treated with bevacizumab (Chi-squared = 6.2305; DF 1; *p* = 0.0126).

**Table 1 cancers-14-02464-t001:** Clinicopathological characteristics of 55 patients.

Variable	*n* (%)/Median [Range]
Age	57 (30–78)
FIGO stage	
I	12
II	2
III	36
IV	5
Histological type	
Serous high-grade	39
Serous low-grade	16
Chemotherapy	
Yes	50
No	5
Neoadjuvant chemotherapy FIGO III/IV	19
Therapy response	
Complete response	41
Partial response	9
Stable disease	2
Progressive disease	3
Bevacizumab therapy in FIGO III/IV	
Yes	27
No	14
Recurrence disease	
Yes	34
No	21
Final status	
Life	33
Death	12

**Table 2 cancers-14-02464-t002:** Main parameters of applied MR sequences *.

Parameter	T2 TSE	T2 TSE Fat-Sat Tra	DW EPI Tra	T2 TIRM	Vibe 3D T1 GRE
Repetition time [ms]	4250	2110	3800	6100	3.05
Echo time [ms]	117	123	73	39	1.13
Flip angle [deg.]	137	150	90	150	10
iPAT factor	-	2	2	-	2
Plane	axial, sagital coronal	axial	axial	axial	axial
Number of signal averages	1	1	4	1	1
Field of view—FOV [mm]	360	360	360	360	360
Rectangular FOV [%]	75, 100, 100	100	75	75	75
Breath-hold	No	No	No	No	No
Resolution (mm)	0.7 × 0.7 × 5	1.4 × 1.4 × 5	B value: 0, 50, 500, 1000, 1500	0.9 × 0.9 × 5	1.7 × 1.3 × 3

* The complete study protocol is provided in Appendix A.

**Table 3 cancers-14-02464-t003:** Comparison of ADC, TTP and Perf. Max. En. between LG EOC and HG EOC tumors.

	*t*-Tests; Grouping: LG/HG. Group 1-LG, Group 2-HG								
	Mean 1	Mean 2	t-Value	df	*p*	Valid N 1	Valid N 2	Std.Dev. 1	Std.Dev. 2
ADC	1151.270	8949.186	4.709557	53	**0.000018**	16	39	2083.173	1724.939
TTP	322.875	3175.641	0.163592	53	0.870675	16	39	968.840	1138.951
Perf.Max En.	257.625	2422.821	0.795021	53	0.430147	16	39	725.000	617.955

**Table 4 cancers-14-02464-t004:** Correlation between Ki67 and VEGF expression and diffusion (mean ADC) and perfusion parameters (TTP and Perf. Max. En.) in primary EOC tumor.

	Correlations Marked Correlations are Significant at *p* < 0.05000						
	Mean	Std.Dv.	r(X,Y)	r^2^		n	*p*
KI67 % ADC ave.	636,364 9694.936	335,335 2163.322	−0.298073	0.088848	−227.334	55	**0.027084**
KI67 % Perfusion TTP	636,364 3191.091	335,335 1083.596	−0.209801	0.044016	−156.214	55	0.124207
KI67 % Perf. Max. En.	636,364 2,467,455	335,335 647.823	0.094630	0.008955	0.69202	55	0.491943
VEGF % ADC ave.	531,373 9517.922	276,760 2152.702	−0.285814	0.081689	−208.779	51	**0.042038**
VEGF % Perfusiom TTP	531,373 3290.196	276,760 1062.822	−0.196590	0.038648	−140.352	51	0.166768
VEGF % Perf. Max. En.	531,373 2511.176	276,760 651.763	0.081618	0.006661	0.57324	51	0.569106

**Table 5 cancers-14-02464-t005:** Correlations between ADC, Perfusion TTP and Perfusion. Max. En. in primary tumor and PFS.

		Correlations Marked Correlations are Significant at *p* < 0.05000					
	Mean	Std.Dv.	r(X,Y)	r^2^	t	n	*p*
ADC ave.	945,3801	2,183,185					
PFS [m]	176,471	104,675	0.036697	0.001347	0.20773	34	0.836758
TTP	3,174,706	1,134,073					
PFS [m]	176,471	104,675	−0.51257	0.262726	−3.37685	34	**0** **.001939**
Perf. Max. En.	2,531,176	658,459					
PFS [m]	176,471	104,675	−0.49284	0.242889	−3.20405	34	**0.00306**

## Data Availability

The data presented in this study are available on request from the corresponding authors.

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
