# Peer review of "The Value of Magnetic Resonance Diffusion-Weighted Imaging and Dynamic Contrast Enhancement in the Diagnosis and Prognosis of Treatment Response in Patients with Epithelial Serous Ovarian Cancer"

_cancers, 2022, doi:10.3390/cancers14102464_

Round 1
Reviewer 1 Report
In the present manuscript, the authors investigated the relationship between the results of MRI imaging and clinical parameters. This study was potentially interesting, but there were several concerns. Detailed comments were described following.
- Patients with stage III high-grade serous ovarian carcinoma have multiple lesions. Did the authors evaluate all measurable lesions?
- The authors showed lower mean ADC values in low-grade serous carcinoma compared to high-grade serous carcinomas. However, a negative correlation was found between ADC values and Ki67 expression, and patients with high ADC values had a favorable prognosis. Clinically, high-grade serous carcinoma is clinically more aggressive than low-grade serous carcinoma. Therefore, the result is confusing.
- The authors performed immunohistochemistry, but no images were shown. How did the authors evaluate the expression of ki67 and VEGF? Moreover, the authors should indicate the representative images.
Author Response
Thank you very much for reviewing our manuscript: “The Value of Magnetic Resonance Diffusion-Weighted Imaging and Dynamic Contrast Enhancement in the Diagnosis and Prognosis of Treatment Response in Patients with Epithelial Serous Ovarian Cancer”, and for your helpful comments and suggestions. We modified the text accordingly.
Reply to point 1.
Obviously, in stage III according to FIGO EOC, there is intraperitoneal dissemination. Long-term measurable changes were assessed in terms of the selection of an appropriate method of initiation of treatment (primary cytoreductive surgery or neoadjuvant chemotherapy). It is known from previous publications that the diffusion restriction in implants is significantly higher than in primary tumors, which was mentioned in the introduction. Intraperitoneal implants are usually homogeneous in their structure and well vascularized. In our research, we focused on a primary tumor, structurally heterogeneous- solid-cystic, often with large areas of necrosis. Such a tumor is more difficult to unambiguously assess the baseline ADC values. In addition, the primary tumor, often as a homogeneous mass, is the only focus of EOC in the early stages, ie FIGO I and II. Focusing on the primary tumor allowed to standardize the results of DWI and DCE in the group of patients with different stages of advancement.
Reply to point 2.
We are sorry, but this is of course our mistake. The study showed that higher average ADC values are characteristic of low-grade EOC and lower average ADC values are characteristic of high-grade EOC. This is confirmed in Table 3, which shows the mean ADC value of 1157.270 for LG EOC, while for EOC HG the mean value of ADC was 894.918. We have already referred to the correct correlations in the discussion. In the version of the work presented after the reviews, this has been corrected both in the main text (results) and in the abstract.
Reply to point 3.
The methods used in the determination of Ki67 and VEGF are presented in the material and method section, in the part immunohistochemistry. The percentage of Ki67 or VEGF staining positive cells was calculated for each patient. As suggested by the reviewer, we attach to the work representative photos with immunohistochemical staining of Ki67 and VEGF from our material.
Thank you for the comments in the review
Paweł Derlatka

Reviewer 2 Report
Thank you for submitting your work and selecting me for reviewing.
The article is comprehensive, robust stat support, and very well done.
My suggestion to further enhance your works would be:
- Include a study workflow, which would be easier for the readers.
- Tables 3 and 4 need to be reformated to fit nicely on the page.
- Please include the limitations of the article in the paragraph before the conclusion
- a tabular presentation of the discussed papers and their limitations/strengths would be helpful for the readers.
- Please mention ADC values in the abstract.
- please mention exclusion criteria, if any
- As a consensus, in a research article, the discussion section starts with a summary of the study results.
- The MRI sequences don't mention separate non-contrast T1 weighted images, in or out-phase images, which is different for pre-contrast fat sat T1 . is it a specials study protocol, because it doesn't follow the international guidelines.
- I think it is important that both the radiologists interpreting the images are board-specialist. Is there any reason for selecting a senior resident? there is a great disparity in the reader, as 1 has 15 years of experience and the other would-be maximum of 2-3 years of MR.
- since all the studies were done on 1.5 T, the author should stat that in the abstract.
- Tables 3, 4, and 5 can be provided as supplementary data supporting the study.
- page 1. citation section last name first name is missing
On a lighter note, the first author is a gynecologist in this radiology-heavy article. Nonetheless, the rest are radiologists.
Round 2
Reviewer 1 Report
The manuscript has been improved.